# Does sex of the jockey influence racehorse physiology and performance

**Charlotte Schrurs**[1]*, **Guillaume Dubois**[2], **Emmanuelle Van Erck-Westergren**[3], **David S. Gardner**[1]*

1 School of Veterinary Medicine & Science, University of Nottingham, Sutton Bonington, Loughborough, United Kingdom, 2 Arioneo, Paris, France, 3 Equine Sports Medicine Practice, Waterloo, Belgium

* charlotte.schrurs@nottingham.ac.uk (CS); david.gardner@nottingham.ac.uk (DSG)

**Data Availability Statement:** All anonymized data used in this manuscript are available from Arioneo, Ltd at https://vet.arioneo.com/en/bdd-publication/ (racehorse training data with sex of jockey listed) and all data used to calculate the win percentage of male or female jockeys are available at The

## Abstract

The racing industry is supported by a predominance of female stablehands and work riders, but few become professional jockeys. Female jockeys have recently had notable race success. No study has assessed whether the sex of the rider may subtly influence racehorse physiology to affect performance. Here, using a validated exercise tracking system (the 'Equimetre'™) that records many physiological parameters simultaneously, this study characterised racehorse cardiovascular (heart rate, heart rate recovery) and biomechanical (stride length and frequency) parameters at various exercise intensities (slow canter to hard gallop) to address the question whether any parameter varied according to sex of the rider. A total of 530 Thoroughbreds, varying in age (2–7 years old) and sex (including geldings), from one racing yard in Australia, completed a total of 3,568 exercise sessions, monitored by a single trainer, on varying track surfaces (sand, turf, or fibre). Different work riders, 103 in total (male, n = 66; female, n = 37) of which n = 43 were current or past registered professional jockeys, participated in the study. Data were analysed using analysis of variation (ANOVA) or mixed-effect models, as appropriate. Sex of the rider did not influence ($P > 0.05$) racehorse speed nor stride length at any training intensity. Racehorse heart rate and peak heart rate increased with training intensity ($P < .001$), with no difference according to sex of rider ($P > 0.05$). Racehorse heart rate recovery was influenced by sex of the rider, but only at the extremes of the reversed, usual training intensity on each surface (e.g. heart rate after galloping on sand was significantly lower with male riders, $P = 0.03$). Finally, analysis of 52,464 race results indicated a similar chance of a top-three placing for male and female jockeys. In conclusion, this study, using objectively obtained data, demonstrates for the first time no overt effect of the rider's sex on racehorse physiology in training and performance in racing. Such data could encourage greater female participation in racing and improve access of female jockeys to better quality mounts in racing events.

## Introduction

In human elite sport, athletes are almost invariably segregated by sex. Male athletes tend to perform at a faster, higher, and stronger level due to a combination of various physical and

University of Nottingham research data repository at http://doi.org/10.17639/nott.7211.

**Funding:** The authors received no specific funding for this work. Arioneo Ltd provided all data used in this study but had no role in study design or analysis, decision to publish, or preparation of the manuscript.

**Competing interests:** The authors received no specific funding for this work. Charlotte Schrurs is self-funding her PhD. All data were collected by Arioneo Ltd. David S Gardner is funded by The School of Veterinary Medicine and Science, University of Nottingham. Guillaume Dubois is an employee of Arioneo Ltd and had no influence on the reporting of results as presented. Emmanuelle Van Erck-Westergren is an Equine Sports Medicine specialist and consultant for Arioneo Ltd. This does not alter our adherence to PLOS ONE policies on sharing data and materials.

morphological advantages [1]. Men usually have greater lean mass, less body fat and greater aerobic and cardiovascular dimensions such as heart size and cardiac output, suggesting a greater capacity for exercise [2]. Interestingly, when male and female athletes were matched according to performance in a timed race, despite the males being taller, heavier, and with higher haemoglobin concentration the sexes did not differ in measures of aerobic performance such as $VO_{2max}$ ($ml·kg^{-1}·min^{-1}$), heart rate (HR), respiratory exchange ratio, or the ventilatory equivalent of oxygen during submaximal running or at maximal exercise [3]. Therefore, when obvious physical differences between males and females are taken into account, performance differences become minimal. Nevertheless, males and females likely differ in various functions unrelated to lean mass that suggest tailoring training programs, equipment or nutrition to optimise performance with these differences in mind would be beneficial [4–6]. In any sport conducted at the highest level, the difference between winning or losing may purely be psychological. Here, men seemingly have an advantage with greater self-confidence [7], less pre-competitive anxiety [8] and an ability to focus purely on performance [9] often being reported. Over the last few decades, representation of women in elite sport has markedly increased. Women were only allowed to officially compete in an Olympic marathon in 1972 (Boston, USA), whereas in Tokyo 2020, medal opportunities for male and female athletes were near parity (49% female representation rate). This represented a rise of 45% since Rio in 2016, and 38% since Sydney in 2000 [10,11]. In various professional sports, female teams are increasingly being created, equality of prize money is being realised, and greater female participation rates are raising the popularity of women's sport [12–15].

In horse-racing, both male and female horses and male and female jockeys compete against each other in most races. An average racehorse weighs ~500-600kg, an average jockey, ~49–55 kg (e.g. see https://www.racing.com/jockeys). Yet, a few 100g extra, on the back of a racehorse, has been shown to influence race performance; thus, additional weights are often used in 'handicap' races, to potentially equalize any performance advantage [16–19]. In other equestrian sports where there is no attempt to control for perceived differences between sexes (both animal and human) such as show-jumping [20,21] or eventing [22], there is no evidence to suggest any influence of the riders' sex on performance. At a recreational level, in most equestrian sports, riders are predominantly female [23–25]. At an elite level, this ratio is often reversed [26]. In horse racing, despite an increase in participation rates for female jockeys, the proportion of female/male jockeys riding at the elite level in racing nations such as England, France and the USA remains low [27] and see https://www.britishhorseracing.com/racing/participants/jockeys/). Physical strength, body shape and tradition were reported by Roberts & MacLean as perceived reasons of restricted opportunities for female jockeys, or the intimidating nature of the weighing room [28]. In 2018, the French racing jurisdiction 'France Galop', implemented a 2kg weight allowance for female jockeys to encourage greater use of female jockeys [29].

Analysis of betting behaviour on races in the UK and Ireland [30], or USA [31], was studied as an indirect proxy for public confidence in the ability of male or female jockeys to win races. In the UK and Ireland, a slight underestimation of the ability of female jockeys to win was found between the years 2003–2013 (i.e. female jockeys won +0.3% more races than estimated from the subjective assessment of the quality of the racehorse) [30]. In the USA, the opposite was found for better quality 'stakes' races (-2.0% [31]. Any successful jockey must possess a combination of strength endurance, balance, fast reaction time and flexibility to be successful [32,33]. Such attributes are equally shared by males and females whom self-select to become successful jockeys, given equal opportunities to ride quality racehorses.

Successful jockeys are thought to readily form a partnership with each racehorse [34]. Competition, and the anxiety associated with it–as reflected in heart rate–usually alter in parallel

(i.e. increase) in both horse and rider. In certain circumstances, for example when psychological aspects (i.e. 'pressure') are further imposed, such as performing in front of a crowd, then heart rate increases in the human, but not horse [35]). It is well known that in human athletes during sporting events, cortisol production increases as a result of heightened anxiety or anticipation of the event, to the benefit of their performance [36]. Similarly, cortisol level in horses is higher during competition [37], compared to training [38], but no difference in equine plasma cortisol was found between race winners and losers [39]. Thus, a multitude of complex factors including anxiety, anticipation or other psychological factors could have an effect on an individual horse's race performance. Males and females *per se* approach competition differently [40], and there is no reason to believe this does not apply to jockeys. Yet, to date, there is a paucity of data on whether sex of the rider has any effect on aspects of racehorse physiology that are important for performance (e.g. the cardiovascular system) or on the actual performance itself in races. Using such data during the routine training of racehorses also mitigates any potential effect of race-day anxiety or other psychological factors.

Hence, the aim of the present study, the first of its kind to date, was to leverage a large database of racehorse training data collected routinely in a single racing stable in Australia using a validated fitness tracking device to investigate whether sex of the rider has an effect on aspects of racehorse speed (slow canter to hard gallop), cardiovascular function (e.g. recovery of heart rate) or locomotion (e.g. stride length) during race-speed training sessions. Furthermore, using publicly-available databases of race outcomes (win, second, third place) from Australia and the UK the study also aimed to determine whether sex of the rider has an effect on racing performance. Our hypothesis was that sex of the rider has no effect on racehorse physiology in training or performance in racing.

## Methods

This retrospective, observational study used two large datasets: 1) a database of racehorse training sessions collected from a single racing yard (Ciaron Maher Racing) in Australia, where sex of the work-rider was known and 2) publicly-available datasets of race results in Australia and in the UK, with sex of the jockey recorded and downloaded from https://www.racing.com/jockeys/ and https://www.britishhorseracing.com/racing/participants/jockeys/.

### Ethics statements

Ethical approval for this study was obtained from the University of Nottingham (School of Veterinary Medicine and Science) Research Ethics Committee (REC code: 3270 201029). The nature of this study involved no experimental protocols. All methods were carried out in accordance with relevant guidelines and regulations. The study is a retrospective, observational study of physiological data held in a database by an external company. The company work with racehorse trainers whom collect data from individual horses that are owned by different individuals. The need for informed consent was waived by the Ethics Committee (School of Veterinary Medicine and Science) due to the retrospective nature of the study, and no personal information was collected on any individual beyond what the data capture device ('Equimetre') records or was inputted as part of routine data collection when exercising racehorses.

**Racehorse physiological outcomes in training according to sex of the jockey. *Horses*.** The study population represented a convenience sample and comprised 530 Thoroughbred racehorses of between 2–7 years of age and included males (colts/stallions; n = 100), females (fillies/mares; n = 262) or geldings (n = 168). Horses were tracked longitudinally as they trained on varying surfaces throughout the year. The racehorses were all regarded by the trainer as race fit; that is, actively in-training to sustain fitness levels and compete in races

during the study period (March 2020 to September 2021). **Riders.** There were 103 different work riders involved in this study. Participants were either regular work-riders who had never registered as a professional race jockey or were either current or past, professionally registered jockeys. The dataset comprised female work riders (n = 37), of which n = 8 were current or past registered professionals on https://www.racing.com/jockeys, n = 29 female riders had never registered as a professional. The remainder were male work riders (n = 66), of which n = 35 were current or previous registered professionals. n = 31 male riders had never registered as a professional. **Equipment.** Horses wore their regular tack and were exercised by a randomly allocated work rider. Rider-horse allocations changed for each individual training session. The device was fitted prior to training by persons accustomed to the device. An electrode was fastened at the girth and the second secured under the saddle pad in the natural dip of the back, as previously described [41,42]. The fitness tracker recorded physiological and locomotory parameters during designated periods of trot, canter and gallop. All trainers and local data analysts had previously integrated the Equimetre™ into their work regimes.

**Training.** The trainer directed each rider to work their respective horse according to given target timings for set distances, determined by a sound and light system fixed to the riders' helmet signalling to them the time taken to cover each furlong (200m). Each training session was evaluated based on the trainer's observation on the track, riders' feedback following exercise and the fitness tracker recordings ('Equimetre'™, Arioneo Ltd, France). Following exercise, all horses were systematically placed on a horse walker or walked in hand to recuperate until heart and respiratory rate had returned to a natural 'baseline'. The training locations were on racetracks or training centres with a choice of surface including sand, turf or fibre. These latter aspects were also recorded for each training session. Using the Equimetre, data were available on speed (i.e. time taken to cover 200m in seconds) recorded for each 200m segment (at 200, 400, 600, 800, 1000, 1200 and 1400m using GNSS (GPS+Glonass+Galileo) satellite data). From these data, the Equimetre also calculated the fastest 200, 400, 600, 800 or 1000m segment. Using the 'fastest 200m' recorded, a data-driven categorisation of speed during the session from slow/medium/hard-canter to slow/medium/hard-gallop was adopted. Descriptive information on each training session was checked, and any sessions with missing information on the jockey was discarded.

**Data collection.** All data collection occurred between March 2020 to September 2021. A total of 3,568 training sessions were available with the rider recorded, with each racehorse completing 5 (2–10), median (first-third interquartile range [IQR]) training sessions. All 530 racehorses completed at least one training session, 134 racehorses completed 10 races, 22 racehorses completed 20 training sessions. Few racehorses completed ≥25 training sessions, therefore these were grouped. Month of training session was recorded for both years. From the exact date of training, the number of training sessions, and the interval between them, could also be recorded for each horse. The Equimetre recorded aspects of each horse's cardiovascular responses to exercise (e.g. heart rate, HR) during trot, canter, gallop, peak heart rate (HRpeak) and HR during recovery (HR at 15 or 30 mins or at the end of the designated training session) and aspects of locomotion (stride length and stride frequency). From these live data that were instantly recorded by the device, further analyses were able to be conducted *post-hoc* that pertained to each training session such as the deltaHR (or rate of early recovery) ([HRpeak–HR at 15min recovery]/15), HR area under the curve (or overall recovery) (HRauc; ([HRpeak +HR at 15min]/2)+([HR at 15min+HR at 30min]/2)+ ([HR at 30min+HR at end of session]/2)). All areas under the curve (AUC) were calculated according to the trapezoid rule in Graphpad Prism (Graphpad Prism 9, Graphpad Software, La Jolla, USA). Environmental temperature and precipitation were recorded as potential covariates in any analyses.

**Racehorse race outcomes according to sex of the jockey.** *Race results.* For Australia, data for the 2021 racing season in Victoria state, involving all race meetings, were extracted online from https://www.racing.com/jockeys/. Aspects of the dataset such as venue, distance, track and class were not taken into consideration. Comparative data in the United Kingdom during the 2021 racing season, exclusively for flat race meetings, were provided by The British Horseracing Authority https://www.britishhorseracing.com/racing/participants/jockeys/. Both datasets included: Australia, n = 169 registered jockeys (male, n = 114, female, n = 55); UK, n = 436 registered jockeys (male, n = 307, female, n = 129). Overall, the combined dataset included a total of 52,464 race starts with all jockeys registered completing at least n = 1 professional race. Recorded data included wins and win percentage (number of wins per total number of starts expressed as a percentage) for individual jockeys and, for Australia only, achieving a podium place (1st, 2nd or 3rd).

**Statistical analysis, *training data*:** Any descriptive data (e.g. continuous variables) that were normally distributed (e.g. speed, stride length, stride frequency) are presented as mean (± 1 standard deviation [SD]). Similar data that were not normally distributed or categorical are presented as median (1st - 3rd interquartile range) or as percentage (of total number) for categorical variables. Data distribution was checked either by standard tests (e.g. Shapiro-Wilk test) or checking of residuals post analysis. If necessary, data were log-transformed ($log_{10}$) to normalise the distribution of the data prior to analysis. For some analyses, where assumptions for analysis of variance could not be met due to missing data (e.g. occasional missed speed or HR recording), then linear mixed models (restricted maximum likelihood; REML) was used, so that any missing data were assumed to be randomly distributed amongst treatment groups. Individual racehorses were included as random effects in the model to account for the reduced within-animal variation. Any available factors that were not part of the design but may influence outcome were included as co-variates (e.g. interval between training session in days, training month, temperature, precipitation). Different training centre/location was also added as further random effects when they had tracks of differing surfaces at the same location. Since few training sessions were conducted on fibre (all-weather), for further analyses of horse-level variation in cardiovascular or locomotory parameters then only sand and turf were considered, and these were analysed separately. **Statistical analysis, *race data*:** The percentage of race wins as a proportion of total races entered was analysed by logistic regression with a logarithm-link function, and sex of the rider as the only fixed effect. Data for both countries were combined to increase sample size, but country was included as a fixed effect for potential secondary outcomes. With the overall win percentage estimated at 10% of total races started, in order to observe a 1% difference between male and female jockeys, with 90% power and 5% significance level would require a total sample size of n = 9781. All data were analysed using Genstat v21 (VSNi Ltd, Rothampsted, Harpenden, UK). Statistical significance was accepted at P<0.05.

## Results

### Descriptive data of the cohort; number, type and intensity of training session by sex of work-rider

Female riders completed a higher proportion of the total number of training sessions in this cohort (n = 1868 of 3568, 52.4%). The number of training sessions ridden by female riders was 19 (4–76) versus 9 (3–23) for male riders. The average distance horses were exercised was not different according to sex of the rider, female, 1981 (1701 – 2155m); male, 1861 (1564–2160) metres, median (IQR). As the speed of the session increased from slow canter to hard gallop, the proportion of male riders increased (Fig 1A). Data for training sessions using an Equimetre

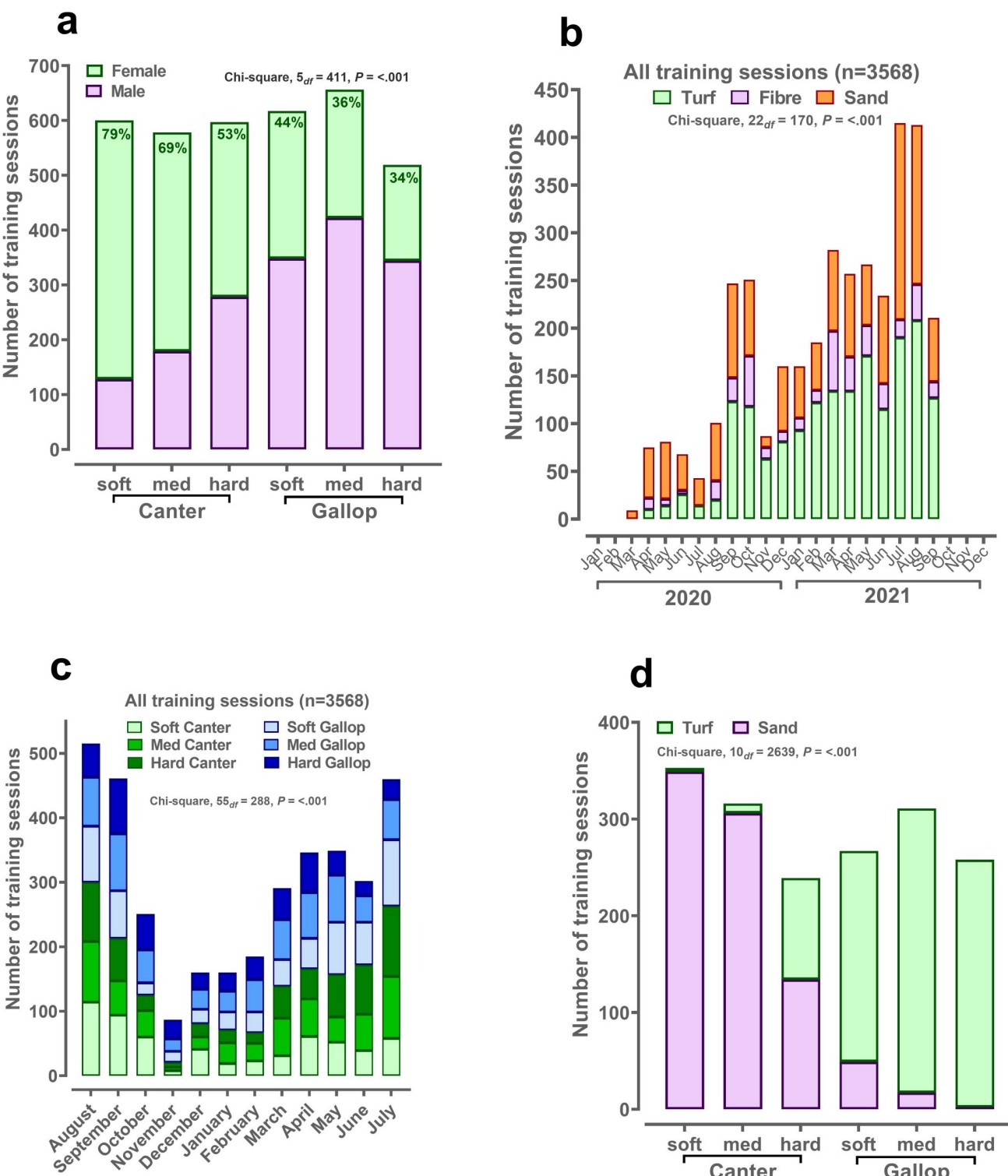

**Fig 1. Classification of the number, type and intensity of training sessions.** Data are numbers of training sessions stratified by training intensity, month of the year or track surface. A total of 3568 training sessions were included in this dataset from a single trainer in Australia. Data were available throughout the year. Training intensity (soft/med/hard canter; soft/med/hard gallop) was derived from calculating sextiles of the fastest furlong (200m interval) for the overall dataset. Statistics were generated by analysing proportions (percentage of group total) by chi-square (Genstat v20, VSNi, Rothampsted, UK).

were available from March 2020 to September 2021. Sessions were conducted on all three surfaces (fibre, turf and sand) throughout the year and gradually increased in frequency, reaching a maximum number of sessions (>400) in July-August 2021 (Fig 1B). Similarly, the number of training sessions stratified by training intensity (i.e. soft canter to hard gallop) followed the pattern of total training sessions over the course of both years with no marked variation in any particular month (Fig 1C). However, when considering the track surface chosen for specific training intensities then it was clear that the majority of canters and gallops were conducted on sand and turf, respectively (Fig 1D). For each track surface, variation in training intensity existed but, for example, only n = 3 (of 1381, 0.2%) hard gallops were conducted on sand, and only n = 28 (of 2165, 1.2%) soft canters were conducted on turf.

## Racehorse speed, stride length and effect of training on differing surfaces by sex of work-rider

Racehorses increase speed by partially increasing stride frequency (by ~ 19% and 20% for horses ridden by female and male riders, respectively) but with a far greater increment in stride length, which increases by ~56% and 57%, respectively; Table 1). The increment in speed with training intensity is not linear; that is, near maximal speeds are achieved at 'soft gallop' with only small increases thereafter (Table 1). The mean time taken for any 200m segment (i.e. each furlong) gradually decreases, the further the training distance, with no effect of rider sex on either turf or sand (Fig 2A and 2B). Stride length also increased with training intensity in a curvilinear fashion on both training surfaces, with no effect of sex of the rider (Table 1, Fig 2C

**Table 1. Descriptive data for the complete cohort of racehorses training at a single racing yard in Australia.**

| Parameter | Rider Sex | Soft Canter | Med Canter | Hard Canter | Soft Gallop | Med Gallop | Hard Gallop | *P*-value *sex* |
|---|---|---|---|---|---|---|---|---|
| Best 200m (secs) | Male | 20.9 ± 2.2 | 15.9 ± 1.2 | 12.9 ± 0.5 | 11.8 ± 0.2 | 11.3 ± 0.1 | 10.8 ± 0.2 | 0.16 |
| | Female | 21.1 ± 2.2 | 16.3 ± 1.2 | 13.1 ± 0.6 | 11.8 ± 0.2 | 11.3 ± 0.1 | 10.8 ± 0.2 | |
| Best 600m (secs) | Male | 65.8 ± 9.5 | 49.7 ± 3.9 | 41.0 ± 2.0 | 37.4 ± 1.1 | 35.6 ± 0.8 | 34.2 ± 0.9 | 0.07 |
| | Female | 66.5 ± 8.3 | 51.0 ± 4.4 | 41.4 ± 2.8 | 37.6 ± 1.1 | 35.9 ± 0.9 | 34.5 ± 1.0 | |
| Max speed (kph) | Male | 35.5 ± 3.5 | 46.1 ± 3.4 | 56.1 ± 2.6 | 61.0 ± 1.0 | 63.7 ± 0.7 | 66.5 ± 1.4 | 0.20 |
| | Female | 35.1 ± 3.4 | 45.0 ± 3.4 | 55.7 ± 2.6 | 61.0 ± 1.1 | 63.6 ± 0.8 | 66.5 ± 1.4 | |
| Peak heart rate (bpm) | Male | 196 ± 24 | 216 ± 20 | 218 ± 15 | 218 ± 18 | 218 ± 20 | 218 ± 17 | 0.29 |
| | Female | 195 ± 24 | 210 ± 22 | 218 ± 16 | 217 ± 17 | 216 ± 12 | 215 ± 13 | |
| Heart rate at 15min (bpm) | Male | 61 ± 11 | 72 ± 15 | 83 ± 15 | 93 ± 18 | 95 ± 19 | 101 ± 18 | 0.89 |
| | Female | 61 ± 11 | 67 ± 14 | 85 ± 17 | 94 ± 18 | 97 ± 16 | 102 ± 16 | |
| HR recovery auc (units) | Male | 165 ± 23 | 188 ± 29 | 203 ± 33 | 217 ± 38 | 216 ± 43 | 218 ± 47 | **0.02** |
| | Female | 168 ± 23 | 184 ± 28 | 212 ± 30 | 223 ± 38 | 226 ± 36 | 229 ± 41 | |
| Stride frequency (strides per sec) | Male | 2.05 ± 0.07 | 2.17 ± 0.09 | 2.28 ± 0.09 | 2.35 ± 0.08 | 2.39 ± 0.08 | 2.44 ± 0.09 | 0.73 |
| | Female | 2.04 ± 0.10 | 2.17 ± 0.10 | 2.29 ± 0.09 | 2.35 ± 0.07 | 2.40 ± 0.08 | 2.45 ± 0.08 | |
| Stride length (meters) | Male | 4.87 ± 0.43 | 5.92 ± 0.36 | 6.87 ± 0.33 | 7.27 ± 0.25 | 7.47 ± 0.25 | 7.62 ± 0.30 | 0.07 |
| | Female | 4.83 ± 0.41 | 5.78 ± 0.38 | 6.77 ± 0.36 | 7.24 ± 0.23 | 7.43 ± 0.23 | 7.61 ± 0.28 | |

Values are Mean ± 1SD for continuous data recorded by 'Equimetre' (n = 530 different racehorses, n = 3568 different training sessions). Data were available throughout the year. Training intensity (soft/med/hard canter; soft/med/hard gallop) was derived from calculating sextiles of the fastest furlong (200m interval) for the overall dataset. Data were analysed by linear mixed models (REML) for the main effect of jockey sex, training type and their pre-specified interaction. Due to multiple training sessions for each racehorse and each rider, their individual ID's were included in the statistical model as nested, random effects. All data analyses were conducted using Genstat v20 (VSNi, UK).

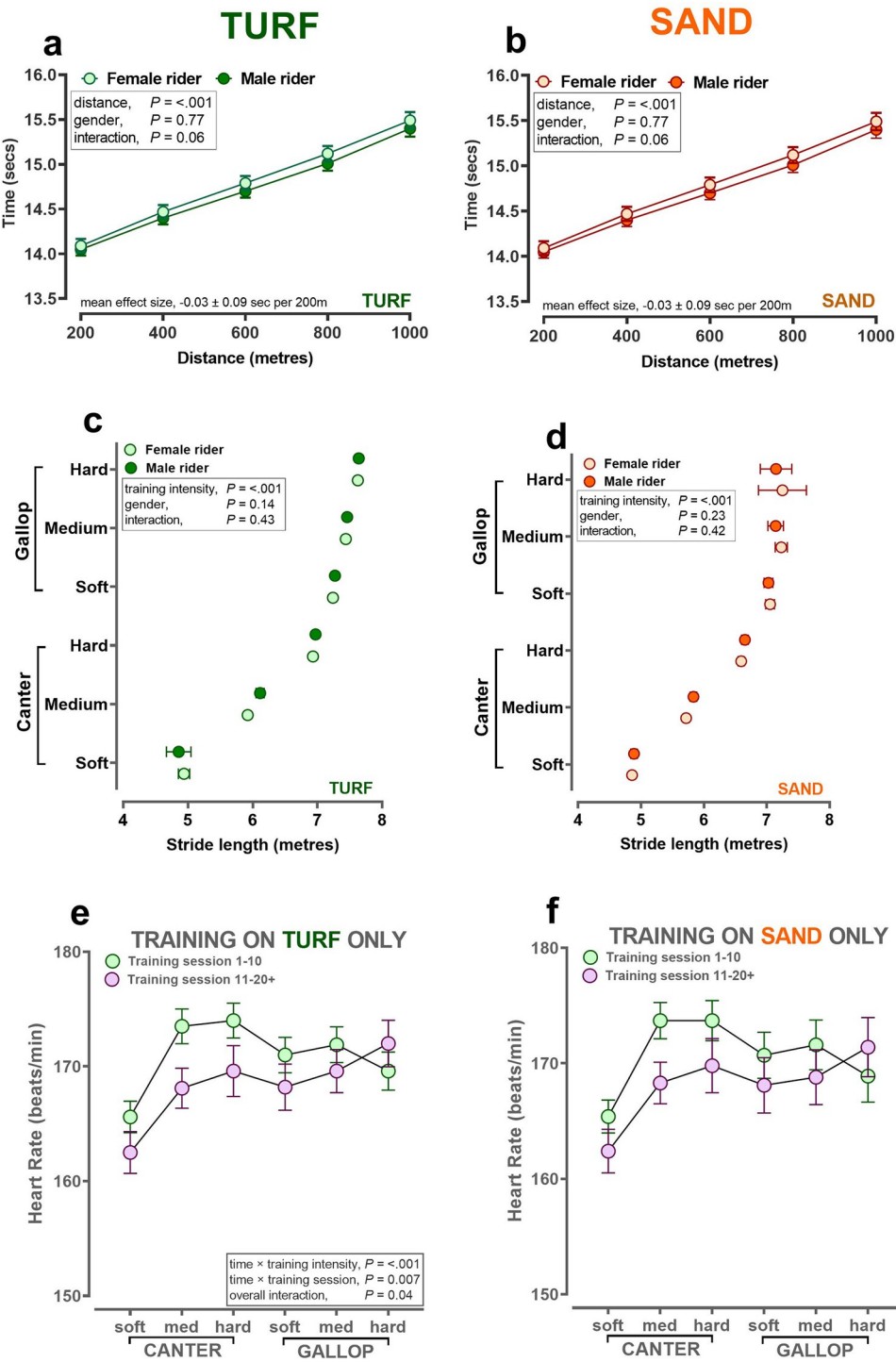

**Fig 2. Speed, stride and influence of multiple training sessions on different track surfaces by sex of work rider. a-f:**
Values are predicted mean ± S.E.M. for continuous data recorded by 'Equimetre' in a cohort of racehorses in Australia
(n = 130 different racehorses, n = 1,754 different training sessions). Data were available throughout the year. Training
intensity (soft/med/hard canter; soft/med/hard gallop) was derived from calculating sextiles of the fastest furlong
(200m interval) for the overall dataset. The statistical model generated predicted means (± S.E.M.) with the pre-
specified interaction, training intensity × rider sex (or training session, **e,f**) fitted last after inclusion of rider
registration status and track surface as fixed effects, HorseID and rider name as random effects, since both horses and
riders completed multiple sessions. All data analyses were conducted using Genstat v20 (VSNi, UK) and graphs
produced using Graphpad Prism v9.0 (La Jolla, USA).

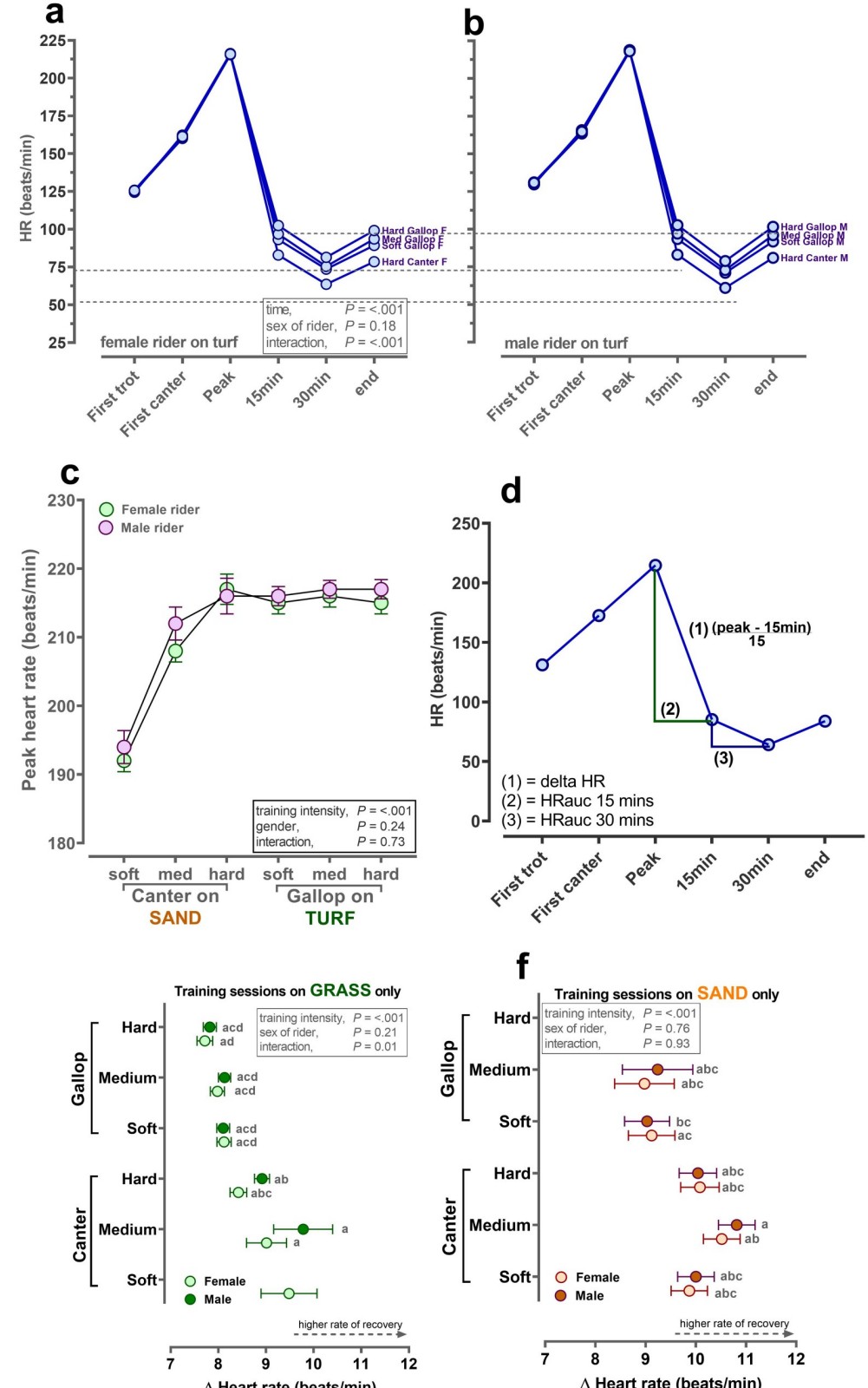

**Fig 3. Racehorse heart rate on different track surfaces by sex of work rider. a,b,c,e,f:** Values are predicted mean ± S.E.M. for continuous data recorded by 'Equimetre' in a cohort of racehorses in Australia (n = 130 different racehorses,

n = 1,754 different training sessions). Data were available throughout the year. Training intensity (soft/med/hard canter; soft/med/hard gallop) was derived from calculating sextiles of the fastest furlong (200m interval) for the overall dataset. The statistical model generated predicted means (± S.E.M.) with the pre-specified interaction, training intensity × rider sex fitted last after inclusion of rider registration status and track surface as fixed effects. HorseID and rider name were included as random effects, since both horses and riders completed multiple sessions. d) describes calculation of delta HR and HR area-under-the-response-curve. All data analyses were conducted using Genstat v20 (VSNi, UK) and graphs produced using Graphpad Prism v9.0 (La Jolla, USA).

and 2D). When training sessions were grouped according to the first 10 conducted versus the next 11–20+, and age of horse was adjusted for, and all heart rates recorded for each horse were analysed in a repeat-measures analysis from trot through canter to peak HR then a small, but statistically significant signal for lower HR during canter and soft-medium gallop on turf (Fig 2E) and sand (Fig 2F) was observed.

## Racehorse cardiovascular responses to incremental training intensity by sex of work-rider

During warm-up, heart rate (HR) for the racehorses in first recorded trot, according to sex of the rider, was: male, 131 ± 1.1 vs. female, 127 ± 1.1 beats/min, $P = 0.02$. During first canter, HR of the horses increased as expected, eliminating any difference by sex of the rider: male, 168 ± 1.4 vs. female, 166 ± 1.4 beats/min, $P = 0.38$ (Fig 3A and 3B). Incorporating heart rate peak, which was not significantly different according to sex of the rider (Fig 3C), with recovery of racehorse heart rate (predominantly over the first 15min after the training session), through 30mins after then end of the session (i.e. the walk back to stables;) allows for the overall area-under-the-response-curve (AUC) to be calculated for the session or for overall recovery (Fig 3D). Over the period of greatest rate of recovery (i.e. to 15mins), the delta HR ([peakHR minus HR at 15mins]/15) appeared influenced by sex of the rider on turf, but not sand (Fig 3E and 3F). That is, for the relatively slower training sessions conducted on turf then horses appeared to recover faster with a male rider. Delta heart rate is primarily driven by the peak heart rate achieved, which itself is influenced by the training intensity. Therefore, this analysis, and the potential signal for an effect of sex of the rider was further drawn out in AUC analyses at 15min recovery (more influenced by peak HR) and from 15 to 30min recovery (less influenced by peak HR).

## Racehorse recovery of heart rate after incremental training intensity by sex of work-rider

Overall racehorse recovery of heart rate, with larger numbers indicating a greater area and thus more beats/min, was influenced by the sex of the rider but *only* at certain training intensities on specific track surfaces (Fig 4A–4D). That is, on turf where the majority of fast gallops are conducted, then racehorses appeared to recover more slowly at 15 and 30min (larger AUC for HR) with a male rider, but the trend was only observed during soft-medium canters i.e. at the paces rarely, but not unusually (n = 119 canters on turf) conducted on that surface (Fig 4A and 4C). In contrast, the opposite was observed on sand; there was a trend for racehorses to recover more quickly with male riders, but only after medium-hard gallops (n = 68 gallops on sand; Fig 4B and 4D). Accumulation of training sessions had little effect on the rate of recovery of heart rate (Fig 4E and 4E). In summary, sex of the rider being male only influenced recovery of racehorse heart rate when horses were exercised at the training intensity rarely conducted on that particular surface e.g. during steady cantering on turf, where most gallops occur and during hard gallops on sand, where the majority of canters were conducted.

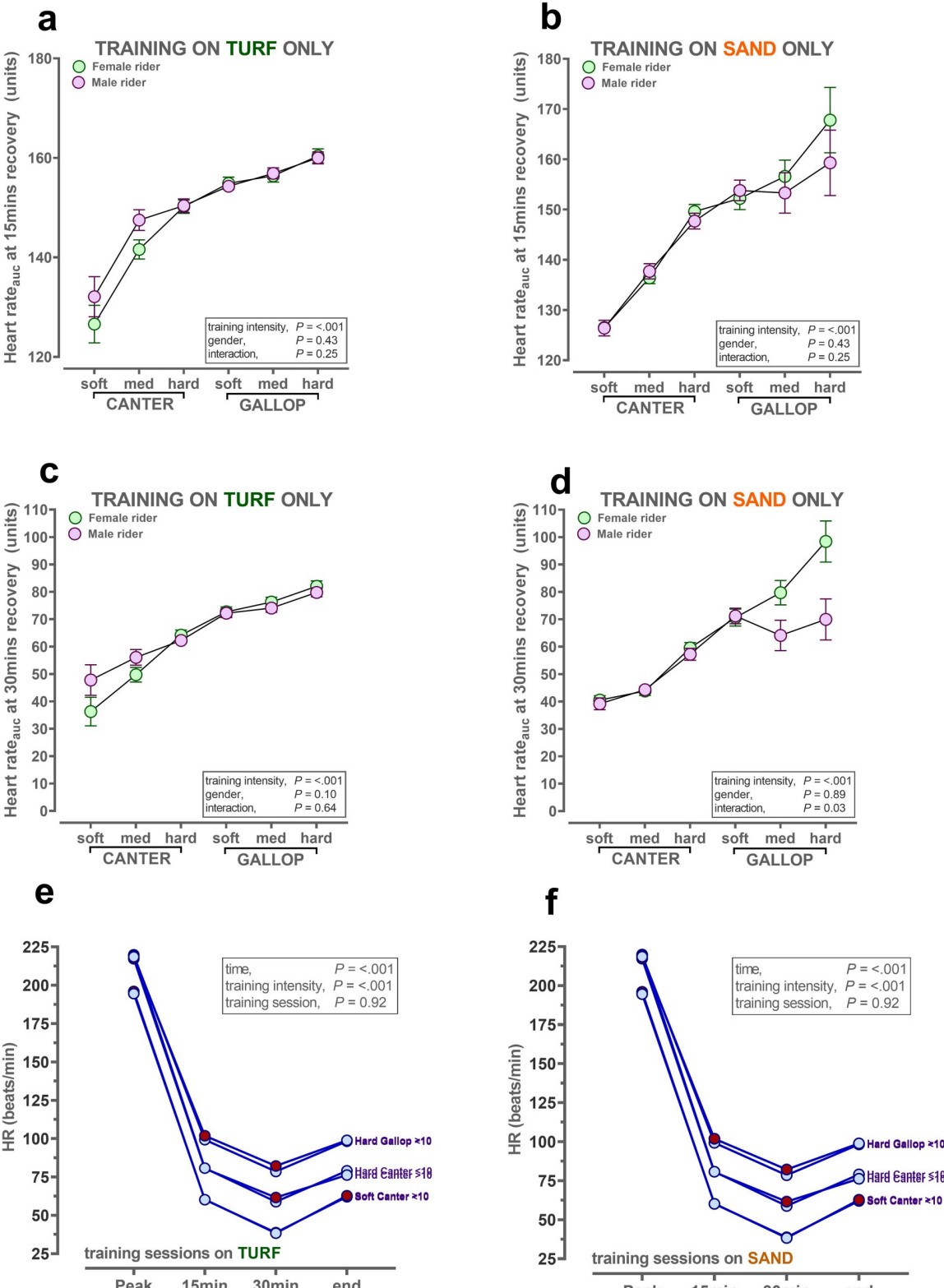

**Fig 4. Recovery of racehorse heart rate on different track surfaces by sex of work rider. a,b,c,d:** Values are predicted mean ± S.E.M. for continuous data recorded by 'Equimetre' in a cohort of racehorses in Australia (n = 130 different racehorses, n = 1,754 different training sessions). Data were available throughout the year. Training intensity (soft/med/hard canter; soft/med/hard gallop) was derived

from calculating sextiles of the fastest furlong (200m interval) for the overall dataset. The statistical model generated predicted means (± S.E.M.) with the pre-specified interaction, training intensity × rider sex fitted last after inclusion of rider registration status and track surface as fixed effects. HorseID and rider name were included as random effects, since both horses and riders completed multiple sessions. HR area-under-the-response-curve (AUC) calculated as described in methods. All data analyses were conducted using Genstat v20 (VSNi, UK) and graphs produced using Graphpad Prism v9.0 (La Jolla, USA).

### Race results

There were far more registered male professional jockeys than female in both UK and Australia. Overall, male jockeys had a small, but significantly greater win percentage compared to female jockeys (female, 9.9 ± 0.5%; male, 11.0 ± 0.2%: F-prob, 0.03). However, the effect was influenced by country; in Australia the effect size was significant (female, 7.9 ± 0.8 vs. male, 10.4 ± 0.4%) but in the UK it was not (female, 10.7 ± 0.7 vs. male, 11.3 ± 0.2%). In Australia, there was no difference between male and female jockeys achieving a top three 'podium' position (female, 25 (10–35) %; male, 27 (18–35) %, Mann-Whitney U test, $P = 0.20$).

### Discussion

No study has directly, and objectively, compared whether female *versus* male jockeys influence any aspect of racehorse physiology (e.g. heart rate) or performance (e.g. speed, stride length). This study, the first of its kind to date, sought to ascertain whether sex of the rider has a measurable difference (positive or negative) on racehorse performance physiology such as speed (e.g. during race-pace training sessions), cardiovascular function (e.g. maximum heart rate, heart rate recovery) or locomotion (e.g. stride length) during training. In addition, we asked whether sex of the jockey has a significant effect on a successful race outcome i.e. either by winning or achieving a top-three placing. There was no overt effect of the sex of the rider on any aspect of racehorse speed or stride length, at race-speed training intensities. However, when the converse training intensity was conducted on the opposite, usual training surface (e.g. a slow canter on turf or a fast gallop on sand, both of which were unusual) then small, but significant physiological effects–reduced or greater rates of recovery of racehorse heart rate, respectively–were observed. Female jockeys had very similar racing success compared to male jockeys. Hence, we suggest that sex of the jockey has few direct effects on racehorse physiology or performance.

In this study, there was a clear difference in the proportion of training sessions completed by male or female work-riders. Despite a lower number of female (n = 37) compared to male riders (n = 64), female riders completed a higher number of training sessions (52.4%) than male riders (47.6%). This contrasts with a previous study, and our current data, where in actual races, there are fewer female riders getting proportionately fewer race rides [43]. Thus, despite female riders completing many more work-rides in training, fewer appear to progress to become professionals, securing race-rides. This observation questions equal racing opportunities between female and male jockeys. Indeed, in training rides, as the intensity of the training session increases, far more male riders are used, with female work-riders completing most slower, canter sessions. This variation may be explained by an unconscious bias made by trainers; the assumption that male riders with greater strength are more suited to higher work intensities. However, our data show that there is no evidence to support this contention; for all training intensities as directed by the trainer, there was no difference in the fastest 200m between racehorses ridden by a male or female work-rider (across 3,568 training sessions).

The faster training sessions may be preferentially ridden by registered professional jockeys of which many more are men. In our dataset, a far greater proportion of 'gallop' sessions were conducted by past or current race registered male professional jockeys (male, 31.8% vs. female,

4.5%). This could, separately to the study, reflect an effect of rider's experience on racehorse performance: trainers choosing to use riders with greater race experience to recapitulate a race environment in training. In Australia, the majority of high-intensity workouts (i.e. gallops) were conducted on turf, while canters were completed on sand. This is consistent with the findings of Morrice-West et al. [44], who surveyed Australian trainers on the use of track surfaces for training. Sand or synthetic 'all-weather' surfaces were commonly used for slow workouts, while gallop work was conducted on turf. It is likely, therefore, that racehorses anticipate certain training intensities according to the track surface. Supporting this contention, in our study, heart rate at trot prior to galloping on grass was 124 ± 26 beats/min (mean ± S.D.) versus 114 ± 26 beats/min prior to a gallop on sand, where most canters occur. Racehorses no doubt anticipate the type of training session they are about to conduct.

To increase speed from canter to gallop, horses increase frequency of their stride, to an extent, but speed predominantly increases due to an extended stride length, as reported here and previously by others [45]. Near maximal speeds (55-60kph) were attained at a training intensity of 'soft-gallop' and only minimally increased thereafter. We acknowledge that speed can be affected by the track condition (i.e. a harder turf or relatively softer sand [46]), which was not measured in this study, but the size of our dataset would limit these relatively small effects. Here, the mean time taken for a horse to cover a furlong (200m), regardless of distance of the training session, did not differ according to the sex of the work rider. As expected, the longer the training session (for example, from 1500m to 3000m), then the average time taken to cover each furlong gradually increased (see Fig 2A and 2B), suggesting a gradual slowing. Interestingly, riding style can influence racehorse speed; a crouched posture reduces aerodynamic drag and can improve racing times by up to 5–7% [47]. Whilst this was not measured in the current study, we assume that male and female work-riders, wearing body protectors, do not adopt significantly different riding postures in training.

Stride length increased with training intensity but was not affected by the work rider's sex. Information on the riders' weights was not obtained for the study. Nevertheless, we appreciate that an increment in the rider's weight has the potential to reduce the stride length of horses and alter performance [48]. The rider's experience, registered race professional *versus* non-registered race professional was accounted for and no marked effect on speed or stride was highlighted. These findings run counter to those of Kapaun et al [49], who found that horses ridden by professional riders had the highest trotting speed and the longest stride length compared to horses mounted by a hobby-rider. The effect of accumulated training sessions on heart rates of horses on both turf and sand, revealed greater fitness levels (i.e. lower heart rate for the same intensity) in horses having completed 11–20+ workouts. Age of the horse was included as a confounder, which couldn't have influenced our results for this observation. Indeed, heart rates during submaximal exercise provide a means of monitoring the adaptation of the cardiovascular system to chronic exercise, commonly referred as the 'fitness" status of an athlete. Foreman drew similar conclusions in Thoroughbreds undergoing exercise testing at different intensities; heart rate was lower following a conventional training program [50].

Racehorse training and racing requires the cooperative effort of two distinctive individuals: horses and humans. Indeed, in any equestrian discipline including racing, positive interaction between horse and rider is paramount to cope with the emotional and physical challenges of the demands of training and the stress of competing. A few studies have described how equine cardiovascular responses interact with the rider; in one, fear or distress signals of the rider were not faithfully transmitted to the horse, as reflected in disparate heart rate responses [51]. However, optimally 'matched' horse-rider combinations translated into reduced HR responses of horses to novel stimuli [52]. Here, there was no effect of sex of the rider on racehorse heart rate, nor the peak of heart rate, during training at differing exercise intensities. That is, when

male and female work-riders are instructed to exercise the racehorses at tempo (canters) or race-pace (gallops) then both do just that–there was no measurable difference, according to sex of the rider–on racehorse speed or on racehorse cardiovascular response. However, it has been shown that rider 'emotion/nerves' can be faithfully transmitted to the horse. For example, when exposed to a novel stimulus [53] or mounted for the first time by a novel rider [54], horses respond with an increase in heart rate. When the rider, but not horse, knows in advance that a novel stimulus/object that is known to cause the horse to startle is about to be encountered, the increment in rider heart rate, in anticipation, is matched by an increment in the horses heart rate [55]. Such acute fight-or-flight responses, most likely mediated by catecholamines and the stress hormone, cortisol, may facilitate improved training responses via energy mobilisation [56] and/or activated behavioural responses [57].

Whilst in this study we found few effects of sex of the rider on the racehorse, we did note a significant effect on recovery of heart rate (see Fig 4), but only at each extreme of exercise intensity when conducted on the track surface that was opposite to that usually used for that exercise intensity; that is, for example, only after medium-hard gallops on sand, heart rate was lower when ridden by a male jockey, whereas after slow canters on grass, heart rate was significantly higher when ridden by a male jockey. Most gallops are done on grass and slow-canters on sand. We interpret this response as indicating that male work-riders, more so than female, may anticipate the 'expected' training-intensity on a given surface and their higher or lower heart rate may be being transmitted through to the horse. Without simultaneous measurement of work-rider heart rate, we cannot confirm this point but it is intriguing and has been noted in other circumstances [58]. Further prospective data collection is warranted to confirm this observation. The extent to which male work-riders 'push' the horses more (i.e. reaching the horses peak HR sooner) on turf, despite instructions to exercise horses at low speed is unknown, but there was no difference in average speed between rider sex at differing training intensities on each surface. It is also possible that for faster sessions on sand (it is unlikely to have happened for slow canters on turf) that changes in work rider for the main galloping session may have provoked changes in the horses' heart rate. Nevertheless, this is only likely to have occurred for faster gallop sessions on sand, where we observed relatively faster recovery of heart rate with male riders aboard. The observations on AUC may be limited by the lack of data on the resting heart rate of the horses. However, this observation was only observed during recovery, where the final recorded heart rate was close to, if not below, usual resting heart rate of race horses.

In addition to the training data, the study also used publicly available race data to investigate whether male or female jockeys have greater competitive success. While small effects were observed in Australia for win percentage (~1% favouring male jockeys), but not for a placing in the top three of each race, no difference was observed in the UK. Therefore, on balance, we suggest that there is also no discernible difference according to sex of the rider on racehorse performance in competitive races. The ~1% difference is likely attributable to factors not recorded in our dataset, such as male riders getting more rides on better horses with a higher rating and therefore more likely to win in the first place. We were not able to get the starting odds for all the races to adjust for the 'quality' of the races. A previous study suggested that women win 0.3% more races than the British and Irish racing markets predict [30], suggesting that, as far as betting behaviour goes, there maybe a slight underestimation of the ability of female jockeys to win races.

The study has a few limitations which should be acknowledged. First, there is a risk of selection bias due to the method of convenience sampling. However, the large sample size should minimise such an effect. Secondly, the study design is retrospective and descriptive. As a result, there are missing data for some riders and horses, along with other variables that have not

been considered such as riders anxiety levels, heart rate responses and weight. Active management of the latter, particularly for professional jockeys could have influenced rider performance, although this is unlikely during routine training sessions as recorded in our study. Significant effects of rapid weight loss (i.e. 2% dehydration), have been noted to reduce performance, for example a 5% reduction of leg strength and 14% reduction in chest strength, with no effect on cognition [59]. Our datasets do not record whether or not any work-rider was to any extent, injured or was carrying any unreported injury, which could impact performance. We have no reason to suspect a substantial bias between males/females with or without injuries in our dataset, despite studies outlining that riders do not necessarily rest or fully recuperate when injured [60]. Nevertheless, using mixed-effect models to analyse the data with due incorporation of any possible hitherto unforeseen confounders should mean that any missing data or confounding factors are distributed at random, minimising any inherent bias in our data.

The number of training sessions available for each individual rider was not balanced; many riders had only participated in <5 training sessions, while some had participated in >100. However, we were not as interested in individual rider effects and incorporating jockeyID in REML models, allows for such unbalanced datasets. When grouped by our main outcome, sex of the jockey, then the proportions of sessions was approximately even overall. Future studies could also include the horse's body weight (if recorded routinely), plasma lactate and cortisol in the racehorses and aspects of the rider's physical status (e.g. heart rate) which could provide novel insight into the unique interactions between different riders and different horses. Further research could expand on performance profiling in both male and female riders across different racing nations and types of racing (e.g. National Hunt or jump-racing).

To conclude, by leveraging a relatively small, but well-controlled dataset, we were able to demonstrate for the first time no overt effect of the rider's sex on racehorse physiology in training and performance in racing. Using measurements from objective fitness tracking systems, no marked sex differentials between work riders were observed on racehorse cardiovascular physiology, locomotory profiles and speed in race-speed training sessions. Furthermore, no marked effect of sex of the rider was noted on hard outcomes such as race performance; the chance of achieving a top three position during >52,000 competitive races. Therefore, this study provides objective evidence that female jockeys are as effective as male jockeys on racehorse physiology and performance. The data should encourage greater female participation in racing and improve access of female jockeys to higher quality racehorses competing in more prestigious races.

## Acknowledgments

The authors would like to thank the trainers, work riders and sports performance analysts at Ciaron Maher Racing for sharing their data, insights and knowledge on the training of their racehorses. This study would not have been possible without their participation. All authors would particularly like to express their appreciation to Romane Borrione.

## Author Contributions

**Conceptualization:** David S. Gardner.

**Data curation:** Guillaume Dubois, David S. Gardner.

**Formal analysis:** Charlotte Schrurs, David S. Gardner.

**Investigation:** Charlotte Schrurs, David S. Gardner.

**Methodology:** David S. Gardner.

**Project administration:** David S. Gardner.

**Supervision:** Emmanuelle Van Erck-Westergren, David S. Gardner.

**Writing – original draft:** Charlotte Schrurs, David S. Gardner.

**Writing – review & editing:** Charlotte Schrurs, Guillaume Dubois, Emmanuelle Van Erck-Westergren, David S. Gardner.

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
