## [Decision Letter · Decision Letter 0]

31 May 2022

PONE-D-22-10266Does sex of the jockey influence racehorse physiology and performancPLOS ONE

Dear Dr. Gardner,

Thank you for submitting your manuscript to PLOS ONE. After careful consideration, we feel that it has merit but does not fully meet PLOS ONE’s publication criteria as it currently stands. Therefore, we invite you to submit a revised version of the manuscript that addresses the points raised during the review process.

We look forward to receiving your revised manuscript.

Kind regards,

Yaodong Gu

Academic Editor

PLOS ONE

Journal Requirements:

 [The authors received no specific funding for this work. Charlotte Schrurs is self-funding her PhD. All data were collected by Arioneo Ltd. David S Gardner is funded by The School of Veterinary Medicine and Science, University of Nottingham. Guillaume Dubois is an employee of Arioneo Ltd. Emmanuelle Van Erck-Westergren is an Equine Sports Medicine specialist and consultant for Arioneo Ltd.]. 

[C.S and D.S.D declare no competing interests exist. Guillaume Dubois is an employee of Arioneo Ltd and Emmanuelle Van Erck-Westergren is an Equine Sports Medicine specialist and consultant for Arioneo Ltd. G.D. and E. V. E-W. had no influence on the reporting of results as presented.] 

Additional Editor Comments:

Reviewers has given some comments for your manuscript. Hope you could revise your manuscript according to these comments, attached it. When you finish the revised

revision and responses to reviewers' comments one by one, especially, the methods part shall more detail.

Reviewers' comments:

Reviewer's Responses to Questions

**Comments to the Author**

1. Is the manuscript technically sound, and do the data support the conclusions?

Reviewer #1: Yes

Reviewer #2: Yes

2. Has the statistical analysis been performed appropriately and rigorously? 

Reviewer #1: Yes

Reviewer #2: Yes

3. Have the authors made all data underlying the findings in their manuscript fully available?

Reviewer #1: Yes

Reviewer #2: Yes

4. Is the manuscript presented in an intelligible fashion and written in standard English?

Reviewer #1: Yes

Reviewer #2: No

5. Review Comments to the Author

Reviewer #1: This manuscript entitled “Does sex of the jockey influence racehorse physiology and performance” primarily aimed to explore the influence of gender differences of the riders on cardiovascular and biomechanical parameters of racehorse. The authors bring an interesting study, but there are still some problems that cannot up this review to a publishing level. Some suggestions are listed in the specific comments below.

Specific comments:

1. In the abstract part, line 43-44, “we objectively report the effect of rider’s sex on racehorse cardiovascular (heart rate, heart rate recovery) …” Please write it in the past tense.

2. In the abstract part, please highlight the implications as well as the meaning of your research.

3. In the introduction part, line 73, “such as VO2max (ml·kg–1·min–1), HR,” please provide the full name of the abbreviation “HR”.

4. In the introduction part, in the opinion of reviewer, the authors provided too much description in this part, which may be too long-winded. I suggest that the authors reduce the first three paragraphs of your introduction to one paragraph.

5. In the methods part, line 274, “All AUC were calculated…” What does the abbreviation “AUC” refer to?

6. In the discussion part, it is recommended to provide a brief description of the aim and main findings in the first paragraph of the manuscript.

7. In the discussion part, line 411-422, you gave too much description on the background of female jockeys which should appear in the introduction part. Since it come to the discussion part, you should discuss the overall findings of your study.

8. In the discussion part, line 427-434, please avoid using “we”, which is too subjective.

9. In the discussion part, line 572-573, “A previous study suggested that women win 0.3% more races than the British and Irish racing markets predict [23].” The finding of this study is different to your study. what is the implication? Can you provide more information?

10. Table 1, it is recommended to bold P-values that are significant.

Reviewer #2: The authors sought to report the effect of rider's sex on racehorse cardiovascular and biomechanical parameters at various exercise intensities, the purposes of this study appear to be specific, but the research novelty and methodology should be further highlighted and elaborated. The authors need to make major revisions to the article. Here below are specific suggestions for this study.

1. Lines 40-42, it is suggested that the authors should briefly introduce the importance of this study rather than state that it was conducted because of limited research on this topic.

2. Lines 46-52, more method details should be presented here.

3. Lines 55-56, “Heart rate recovery after exercise appeared influenced by sex of the rider”, did this result reach the significant level?

4. Lines 58-60, some specified conclusions should be given here.

5. Please modify and improve the quality of the keywords as this will assist others when they are searching for information on your research topic.

6. Introduction, I could see the gap that the authors were going to bridge. However, I did not see the significance properly. The authors should highlight the novelty of this study. Moreover, this part is too lengthy, the authors should consider shorting the contents. In addition, the authors are suggested to add more previous studies in terms of the gender effects on various kinds of exercise here as a background. Here below are some recommendations.

[1] Effect of gender and running experience on lower limb biomechanics following 5 km barefoot running. Sports Biomechanics, 2021, 1-14.

[2] Gender differences in hemodynamic regulation and cardiovascular adaptations to dynamic exercise. Current cardiology reviews, 2020, 16(1), 65-72.

[3] Comparison of biomechanical characteristics between male and female elite fast bowlers. Journal of Sports Sciences, 2019, 37(6), 665-670.

7. Line 75, “(p>.05)”, not necessary.

8. Lines 201-203, “included males (colts/stallions; n= 32), females (fillies/mares; n=59) or geldings (n=39).”, which in total did not equal 530, please add more explanation.

9. Lines 199-213, a sub-title should be added for this paragraph. Also, it is suggested that the authors should consider presenting the basic information about the racehorses and riders using a table.

10. Lines 228-229, lines 234-235, it is suggested that a detailed figure should be added here for a clearer explanation.

11. Lines 271-274, why did the authors measure these parameters?

12. Some unnecessary contents should be deleted, for example, lines 276-277, “Final datasets were cleaned and checked for artifacts in MS Excel.”.

13. Statistical analysis, the authors should present more details about these contents “Repeated training sessions for each horse were analyzed, if possible, using repeated measures analysis of variance”.

14. The results are duplicates in the results session, on the tables, and also in the discussion session. I recommend the authors keep just the table on the results table and make the discussion stronger. The text on the results session with the description of the table is not helping the reader.

15. Lines 411-434, it is suggested that for the first paragraph, the authors should present the main purposes and findings of this study and indicate if the results were consistent with the hypotheses.

16. The conclusions should be further strengthened based on the findings of this study.

6. PLOS authors have the option to publish the peer review history of their article (what does this mean?). If published, this will include your full peer review and any attached files.

Reviewer #1: No

Reviewer #2: **Yes: **Song Yang

---

## [Author Response · Author response to Decision Letter 0]

22 Jun 2022

1. Style templates

I believe the formatting requirements have been met

2. Financial disclosures

The authors received no specific funding for this work. Arioneo Ltd provided all data used in this study but had no role in study design or analysis, decision to publish, or preparation of the manuscript.

3. Competing interests

Guillaume Dubois is an employee of Arioneo Ltd. Emmanuelle Van Erck-Westergren is an Equine Sports Medicine specialist and consultant for Arioneo Ltd. This does not alter our adherence to PLOS ONE policies on sharing data and materials.

4. Data availability

All anonymized data used in this manuscript (two excel files) are available at the research data repository managed by The University of Nottingham at xxx or are available upon request from the partner company Arioneo Ltd at xxxx

Editors comments

Reviewers has given some comments for your manuscript. Hope you could revise your manuscript according to these comments, attached it. When you finish the revised revision and responses to reviewers' comments one by one, especially, the methods part shall more detail

Comments addressed individually below. 

Reviewer #1 comments

This manuscript entitled “Does sex of the jockey influence racehorse physiology and performance” primarily aimed to explore the influence of gender differences of the riders on cardiovascular and biomechanical parameters of racehorse. The authors bring an interesting study, but there are still some problems that cannot up this review to a publishing level. Some suggestions are listed in the specific comments below.

Specific comments:

1.In the abstract part, line 43-44, “we objectively report the effect of rider’s sex on racehorse cardiovascular (heart rate, heart rate recovery) …” Please write it in the past tense.

Done, line 43-46.

2. In the abstract part, please highlight the implications as well as the meaning of your research.

Included (line 60-63), but bearing in mind abstract word limit.

3. In the introduction part, line 73, “such as VO2max (ml·kg–1·min–1), HR,” please provide the full name of the abbreviation “HR”.

Done, line 76

4. In the introduction part, in the opinion of reviewer, the authors provided too much description in this part, which may be too long-winded. I suggest that the authors reduce the first three paragraphs of your introduction to one paragraph.

The introduction has been reduced from 1502 words to 985 words

5. In the methods part, line 274, “All AUC were calculated…” What does the abbreviation “AUC” refer to? 

Added, line 248

6. In the discussion part, it is recommended to provide a brief description of the aim and main findings in the first paragraph of the manuscript.

Added, lines 388-396.

7. In the discussion part, line 411-422, you gave too much description on the background of female jockeys which should appear in the introduction part. Since it come to the discussion part, you should discuss the overall findings of your study.

Added.

8. In the discussion part, line 427-434, please avoid using “we”, which is too subjective.

Altered.

9. In the discussion part, line 572-573, “A previous study suggested that women win 0.3% more races than the British and Irish racing markets predict [23].” The finding of this study is different to your study. what is the implication? Can you provide more information?

Added this information, line 551.

10. Table 1, it is recommended to bold P-values that are significant.

Bold p-values <0.05 has been done

Reviewer #2 The authors sought to report the effect of rider's sex on racehorse cardiovascular and biomechanical parameters at various exercise intensities, the purposes of this study appear to be specific, but the research novelty and methodology should be further highlighted and elaborated. The authors need to make major revisions to the article. Here below are specific suggestions for this study.

 1. Lines 40-42, it is suggested that the authors should briefly introduce the importance of this study rather than state that it was conducted because of limited research on this topic.

Altered, lines 40-44.

 2. Lines 46-52, more method details should be presented here.

Added, lines 46-48.

 3. Lines 55-56, “Heart rate recovery after exercise appeared influenced by sex of the rider”, did this result reach the significant level?

Added, line 59. This relates to the final figure, and is rather complicated but the P-value is significant for the interaction effect on sand. The next text should explain the effect.

 4. Lines 58-60, some specified conclusions should be given here.

Added, lines 61-64. I believe these conclusions are justified.

 5. Please modify and improve the quality of the keywords as this will assist others when they are searching for information on your research topic.

Keywords as originally proposed seem to be appropriate. Two new keywords added. Any other suggestions are welcome.

 6. Introduction, I could see the gap that the authors were going to bridge. However, I did not see the significance properly. The authors should highlight the novelty of this study. Moreover, this part is too lengthy, the authors should consider shorting the contents. In addition, the authors are suggested to add more previous studies in terms of the gender effects on various kinds of exercise here as a background. Here below are some recommendations.

 [1] Effect of gender and running experience on lower limb biomechanics following 5 km barefoot running. Sports Biomechanics, 2021, 1-14.

 [2] Gender differences in hemodynamic regulation and cardiovascular adaptations to dynamic exercise. Current cardiology reviews, 2020, 16(1), 65-72.

 [3] Comparison of biomechanical characteristics between male and female elite fast bowlers. Journal of Sports Sciences, 2019, 37(6), 665-670.

The introduction has been reduced from 1502 words to 985 words. References as suggested added to introduction at an appropriate place.

 7. Line 75, “(p>.05)”, not necessary.

Removed.

 8. Lines 201-203, “included males (colts/stallions; n= 32), females (fillies/mares; n=59) or geldings (n=39).”, which in total did not equal 530, please add more explanation.

Yes, sorry – that was a mistake. Have rechecked and revised the numbers. Line 180. 

 9. Lines 199-213, a sub-title should be added for this paragraph. Also, it is suggested that the authors should consider presenting the basic information about the racehorses and riders using a table.

Sub-titles added throughout methods. We feel the descriptive information is fine as is, but if the editor would like a table, we can add one.

 10. Lines 228-229, lines 234-235, it is suggested that a detailed figure should be added here for a clearer explanation.

We have referenced previous studies that have included a detailed figure of the device and more technical details that are beyond the scope of this manuscript. Lines 191-194. We could always add one to a new supplementary material, should the editor require.

 11. Lines 271-274, why did the authors measure these parameters?

As described in the manuscript (lines 235 – 238) these parameters were calculated in order to make metric certain aspects of the racehorses carviovascular response such as rate of early recovery (deltaHR) or HR area under the curve (overall recovery metric).

 12. Some unnecessary contents should be deleted, for example, lines 276-277, “Final datasets were cleaned and checked for artifacts in MS Excel.”.

Removed.

 13. Statistical analysis, the authors should present more details about these contents “Repeated training sessions for each horse were analyzed, if possible, using repeated measures analysis of variance”.

The statistics section has been trimmed to better represent the analyses conducted. For example, no repeat-measures of time was conducted when horseID was included as a random effect in the model to account for multiple training sessions (recorded as 1 to n session) on the same horse. A power calc has also been included. Lines 253 – 280.

 14. The results are duplicates in the results session, on the tables, and also in the discussion session. I recommend the authors keep just the table on the results table and make the discussion stronger. The text on the results session with the description of the table is not helping the reader.

We are uncertain as to what is unclear in regard to this statement; the text in the results section describes briefly the results found, indicating which table or figure the description is referring to. 

 15. Lines 411-434, it is suggested that for the first paragraph, the authors should present the main purposes and findings of this study and indicate if the results were consistent with the hypotheses.

Noted, this has now been done. Lines 366-381.

 16. The conclusions should be further strengthened based on the findings of this study.

We feel we have now strengthened the conclusions after the changes have been made.

---

## [Decision Letter · Decision Letter 1]

22 Jul 2022

PONE-D-22-10266R1Does sex of the jockey influence racehorse physiology and performancePLOS ONE

Dear Dr. Gardner,

Thank you for submitting your manuscript to PLOS ONE. After careful consideration, we feel that it has merit but does not fully meet PLOS ONE’s publication criteria as it currently stands. Therefore, we invite you to submit a revised version of the manuscript that addresses the points raised during the review process.

We look forward to receiving your revised manuscript.

Kind regards,

Yaodong Gu

Academic Editor

PLOS ONE

Journal Requirements:

Reviewers' comments:

Reviewer's Responses to Questions

**Comments to the Author**

1. If the authors have adequately addressed your comments raised in a previous round of review and you feel that this manuscript is now acceptable for publication, you may indicate that here to bypass the “Comments to the Author” section, enter your conflict of interest statement in the “Confidential to Editor” section, and submit your "Accept" recommendation.

Reviewer #1: All comments have been addressed

Reviewer #2: (No Response)

2. Is the manuscript technically sound, and do the data support the conclusions?

Reviewer #1: Yes

Reviewer #2: Yes

3. Has the statistical analysis been performed appropriately and rigorously? 

Reviewer #1: Yes

Reviewer #2: Yes

4. Have the authors made all data underlying the findings in their manuscript fully available?

Reviewer #1: Yes

Reviewer #2: Yes

5. Is the manuscript presented in an intelligible fashion and written in standard English?

Reviewer #1: Yes

Reviewer #2: Yes

6. Review Comments to the Author

Reviewer #1: The authors are to be commended for their significant work in revising this manuscript. There are substantial changes that address most of the comments, with great improvement in scope and clarity. However, there are still few issues need to be noticed, especially in the introduction part. Some suggestions are listed in the specific comments below

Specific comments:

1. In the last paragraph of introduction part, please mention the aim of this study.

2. In the conclusion part, line 549-558, “We hope that the results of the study…to better quality mounts in racing events.” It is recommended to improve the logic of these sentences. Findings of this manuscript should be put before the implication and highlights.

Reviewer #2: Most of the previous concerns were addressed in the revised manuscript. However, after further examination, there are still some small issues that need to be addressed before bringing this up to a publishable standard, please find them below.

1. Lines 54-55, 57, it is suggested that the authors could change it to “p > 0.05” since there is no significance.

2. In terms of my previous comment “The results are duplicates in the results session, on the tables, and also in the discussion session. I recommend the authors keep just the table on the results table and make the discussion stronger. The text on the results session with the description of the table is not helping the reader.”, the results section describes the findings of this paper in considerable detail, while these details have already been included in the tables and figures. Therefore, the author may consider making some deletions while keeping the main results of this paper here.

7. PLOS authors have the option to publish the peer review history of their article (what does this mean?). If published, this will include your full peer review and any attached files.

Reviewer #1: **Yes: **Peimin Yu

Reviewer #2: **Yes: **Song Yang

---

## [Author Response · Author response to Decision Letter 1]

2 Aug 2022

Reviewer #1: The authors are to be commended for their significant work in revising this manuscript. There are substantial changes that address most of the comments, with great improvement in scope and clarity. However, there are still few issues need to be noticed, especially in the introduction part. Some suggestions are listed in the specific comments below

Specific comments:

1. In the last paragraph of introduction part, please mention the aim of this study.

Page 4, lines 144-153. A re-written aim/hypothesis has been included, replacing the previous clearly stated hypothesis.

2. In the conclusion part, line 549-558, “We hope that the results of the study…to better quality mounts in racing events.” It is recommended to improve the logic of these sentences. Findings of this manuscript should be put before the implication and highlights.

Page 14, lines 542-553. A re-written concluding paragraph has been included

Reviewer #2: Most of the previous concerns were addressed in the revised manuscript. However, after further examination, there are still some small issues that need to be addressed before bringing this up to a publishable standard, please find them below.

1. Lines 54-55, 57, it is suggested that the authors could change it to “p > 0.05” since there is no significance.

Page 2, lines 54-55, 57. Revision made.

2. In terms of my previous comment “The results are duplicates in the results session, on the tables, and also in the discussion session. I recommend the authors keep just the table on the results table and make the discussion stronger. The text on the results session with the description of the table is not helping the reader.”, the results section describes the findings of this paper in considerable detail, while these details have already been included in the tables and figures. Therefore, the author may consider making some deletions while keeping the main results of this paper here.

Page 8 -, the authors are uncertain what specific changes are required here. The legends of each table and graph help the reader understand that data. In the results section, a summary description is given of each figure or table and the exact data is not repeated as either text/table/graph. Any results in the discussion are specifically incorporated there to help elucidate that discussion point, meaning the manuscript has more of a narrative flow. I do not think ‘making the discussion stronger’ is valid, as essentially any elements of a discussion are the authors subjective comment on their objectively acquired and reported results. I believe this is what is reported here.

---

## [Decision Letter · Decision Letter 2]

8 Aug 2022

Does sex of the jockey influence racehorse physiology and performance

PONE-D-22-10266R2

Dear Dr. Gardner,

We’re pleased to inform you that your manuscript has been judged scientifically suitable for publication and will be formally accepted for publication once it meets all outstanding technical requirements.

Kind regards,

Yaodong Gu

Academic Editor

PLOS ONE

Reviewers' comments:

Reviewer's Responses to Questions

**Comments to the Author**

1. If the authors have adequately addressed your comments raised in a previous round of review and you feel that this manuscript is now acceptable for publication, you may indicate that here to bypass the “Comments to the Author” section, enter your conflict of interest statement in the “Confidential to Editor” section, and submit your "Accept" recommendation.

Reviewer #1: All comments have been addressed

Reviewer #2: All comments have been addressed

2. Is the manuscript technically sound, and do the data support the conclusions?

Reviewer #1: Yes

Reviewer #2: (No Response)

3. Has the statistical analysis been performed appropriately and rigorously? 

Reviewer #1: Yes

Reviewer #2: (No Response)

4. Have the authors made all data underlying the findings in their manuscript fully available?

Reviewer #1: Yes

Reviewer #2: (No Response)

5. Is the manuscript presented in an intelligible fashion and written in standard English?

Reviewer #1: Yes

Reviewer #2: (No Response)

6. Review Comments to the Author

Reviewer #1: (No Response)

Reviewer #2: (No Response)

7. PLOS authors have the option to publish the peer review history of their article (what does this mean?). If published, this will include your full peer review and any attached files.

Reviewer #1: No

Reviewer #2: **Yes: **Yang Song

---

## [Editor Report · Acceptance letter]

10 Aug 2022

PONE-D-22-10266R2 

Does sex of the jockey influence racehorse physiology and performance. 

Dear Dr. Gardner:

I'm pleased to inform you that your manuscript has been deemed suitable for publication in PLOS ONE. Congratulations! Your manuscript is now with our production department. 

Kind regards, 

on behalf of

Professor Yaodong Gu 

Academic Editor

PLOS ONE